# Sex-Sparing Robot-Assisted Radical Cystectomy with Intracorporeal Padua Ileal Neobladder in Female: Surgical Technique, Perioperative, Oncologic and Functional Outcomes

**DOI:** 10.3390/jcm9020577

**Published:** 2020-02-20

**Authors:** Gabriele Tuderti, Riccardo Mastroianni, Simone Flammia, Mariaconsiglia Ferriero, Costantino Leonardo, Umberto Anceschi, Aldo Brassetti, Salvatore Guaglianone, Michele Gallucci, Giuseppe Simone

**Affiliations:** 1“Regina Elena” National Cancer Institute, Department of Urology, 00100 Rome, Italy; riccardomastroianniroma@gmail.com (R.M.); marilia.ferriero@gmail.com (M.F.); umberto.anceschi@gmail.com (U.A.); aldo.brassetti@gmail.com (A.B.); salvatore.guaglianone@ifo.gov.it (S.G.); puldet@gmail.com (G.S.); 2“Sapienza” University of Rome, Department of Urology, 00100 Rome, Italy; roccosimone92@gmail.com (S.F.); costantino.leonardo@uniroma1.it (C.L.); michele.gallucci@ifo.gov.it (M.G.)

**Keywords:** bladder cancer, female, intracorporeal neobladder, outcomes, radical cystectomy, robotic, sex-sparing

## Abstract

Our aim was to illustrate our technique of sex-sparing (SS)-robot-assisted radical cystectomy (RARC) in female patients receiving an intracorporeal neobladder (iN). From January 2013 to June 2018, 11 female patients underwent SS-RARC-iN at a single tertiary referral center. Inclusion criteria were a cT ≤ 2 N0 M0 bladder tumor at baseline imaging (CT or MRI) and an absence of tumors in the bladder neck, trigone and urethra at TURB. Baseline, perioperative, and outcomes at one year were reported. The median operative time was 255 min and the median hospital stay was seven days. Low-grade Clavien complications occurred in four patients (36.3%), while high-grade complications were not observed in any. Seven patients (63.7%) had an organ-confined disease at the pathologic specimen; nodal involvement and positive surgical margins were not detected in any of the cases. At a median follow-up of 28 months (IQR 14–51), no patients developed new onset of chronic kidney disease stage 3b. After one year, daytime and nighttime continence rates were 90.9% and 86.4% respectively. Quality of life as well as physical and emotional functioning improved significantly over time (all *p* ≤ 0.04), while urinary symptoms and sexual function worsened at three months with a significant recovery taking place at one year (all *p* ≤ 0.04). Overall, 8 out of 11 patients (72.7%) were sexually active at the 12-month evaluation. In select female patients, SS-RARC-iN is an oncologically sound procedure associated with favorable perioperative and functional outcomes.

## 1. Introduction

Radical cystectomy (RC) with urinary diversion is the standard treatment for patients with muscle-invasive (MI) and high-risk non-muscle-invasive (NMI) urothelial carcinoma of the bladder and can offer an orthotopic neobladder (ON) diversion if technically and oncologically feasible [1].

Although bladder cancer (BCa) is more frequent among men, it remains the 17th most common cancer in women worldwide, with approximately 74,000 new diagnosed cases each year [2]. Moreover, women present an advanced stage at diagnosis more often, increasing the requirement of RC [3].

In female patients, the standard surgical procedure is represented by anterior pelvic exenteration including the removal of the bladder, ovaries, uterus, and anterior vaginal wall [1].

In this setting, when an ON is performed, the procedure can be associated with a considerable rate of voiding symptoms [4,5]. In addition, sexual dysfunction derived from such a highly demolitive surgical procedure is a key concern, especially in younger patients due to a significant impact on health-related quality of life (HRQoL) [6].

The improvement of imaging modalities, an increased knowledge of pelvic structure anatomy and function, and an advancement of surgical techniques have enabled less-destructive methods for treating high-risk BCa.

In this scenario, various types of pelvic-organ-preserving techniques, usually named “sex-sparing”, have been proposed [5], aiming at the preservation of neurovascular bundles, vagina, and uterus, combining these techniques in order to optimize sexual and functional results without compromising oncological outcomes. 

Functional outcomes of sex-sparing (sex) RC are essentially based on surgical dissection planes, with the sex-sparing approach being associated with the preservation of utero-vaginal hypogastric plexus, while during standard RC only rectal hypogastric plexus is preserved [7].

The Bern group were the first to describe the feasibility of nerve-sparing RC and ON replacement in female patients, highlighting the potential advantages which derive from preserving pelvic reproductive organs and their nervous structures both in terms of continence and urinary retention [7].

However, a recent systematic review aiming at the evaluation of the advantages and disadvantages of sexual-function-preserving RC and ON in female patients underlined the need for further and more robust comparisons between sex and standard RC as existing data are still immature [5].

Notwithstanding, for well-selected patients, sparing female reproductive organs during RC can be an oncologically safe procedure and can provide improved functional outcomes.

Accordingly, despite the widespread use of robot-assisted radical cystectomy (RARC), there is a paucity of data concerning outcomes of sex-RARC with intracorporeal ON (iON) performed in female patients.

In this paper we describe surgical steps of sex-RARC in female patients, highlighting differences with the standard technique and anatomical details of preservation of the inferior hypogastric plexus (IHP) and we report perioperative, pathologic, and functional outcomes.

## 2. Experimental Section

### 2.1. Patients

Our single-center Institutional-Review-Board-approved BCa database was queried for “Female”, “RARC”, “iON”, and “Sex-sparing”. Overall, 11 patients were treated between January 2013 and June 2018, with a minimum one year of follow-up. Inclusion criteria were a cT ≤ 2 N0 M0 bladder tumor at baseline imaging (CT or MRI) and an absence of tumors in the bladder neck, trigone, and urethra at transurethral resection of the bladder tumor (TURB). Exclusion criteria included any contraindication to ON. All subjects gave their written informed consent for inclusion before they participated in the study.

### 2.2. Surgical Technique

#### 2.2.1. Sex-Sparing Robot-Assisted Radical Cystectomy

The patient was placed in a steep Trendelenburg position, and a six trocars access was performed as previously described [8].

Sex-RARC was performed replicating the principles of open technique described by Bhatta Dhar et al. [7]. After an incision of the posterior peritoneum up to the round ligament, the ureters were identified and meticulously isolated with a “no-touch” technique. The umbilical artery, uterine artery, superior and inferior vesical arteries, and vaginal branches were carefully prepared bilaterally.

Because the uterus was going to be spared, the peritoneum was incised at the level of the utero-vesical junction in order to deflect the uterus and develop a vesico-vaginal plane between the bladder and the anterior wall of the uterus. The vaginal wall dissection at the cervical level was performed in the anterior plane of the vagina at the 2 and 10 o’clock position in order to preserve the utero-vaginal and pararectal components of the IHP (highlighted in red and green colors respectively, in the video), while in the standard technique the dissection is usually performed dorsolaterally at the 4 and 8 o’clock position, preserving only the pararectal plexus and removing en bloc with the specimen and the utero-vaginal components of the IHP. The superior and inferior vesical arteries and veins were secured with Hem-o-lok clips (Teleflex, Wayne, PA, USA) and transected with LigaSure at their origin from the internal iliac vessels, while the uterine arteries and the vaginal branches directed to the paravaginal tissue were preserved. Both ureters were divided between Weck clips, and margins were sent for frozen sections. Next, the Retzius space was approached. Endopelvic fascia was incised very close to the bladder neck in order to reduce the risk of an accidental injury of neurovascular paraurethral structures, which is crucial for both sexual and continence functionality. The urethra was prepared and a sample was sent for frozen section. Bladder was secured in an endobag and extracted through a 3-cm prepubic incision.

#### 2.2.2. Pelvic Lymph Node Dissection and Intracorporeal Orthotopic Neobladder

A meticulous separate package extending pelvic lymph node dissection (PLND) was performed, including obturator, internal, external, and common iliac nodes. Considering that superior hypogastric plexus (SHP) is usually located just below the aortic bifurcation, ventrally to the sacral promontory, presacral nodes are not removed. Moreover, lymphatic tissue medial to internal iliac arteries which is in close contact with uterine and vaginal vessels and with uterine and vaginal plexus is usually spared.

After RC and PLND, intracorporeal Padua ileal neobladder was performed as previously described [8].

### 2.3. Outcomes Evaluated

Collected demographic parameters were age, body mass index (BMI), gender, and American Society of Anesthesiologists (ASA) score. Clinical variables were preoperative eGFR, preoperative hemoglobin (Hgb), and neoadjuvant chemotherapy rate. Surgical outcomes reported consisted of operative time, Hgb at discharge, hospital stay, and complications according to the Clavien–Dindo system [9]. Pathological findings including pT stage, pN stage, histology, lymph node count, and the positive surgical margin status were analyzed. Functional outcomes assessed were the last eGFR, neobladder stones rate, the uretero-ileal strictures rate, and the need for intermittent self-catheterization. Daytime and nighttime continence recovery probabilities were assessed over time. EORTC QLQ-C30 and EORTC QLQ-BLM30 questionnaires were adopted to assess HRQoL and urinary symptoms respectively. Every item measured ranged in a score from 0 to 100. A Female Sexual Function Index (FSFI) questionnaire was adopted for sexual function assessment [10]. Each of the six sexual domains range in score from 0 to 6, with a maximum global score of 36. Questionnaires were administered at baseline, and at 3 and 12 month follow-up.

As supplementary data, we reported preoperative perioperative, pathologic and functional characteristics comparisons of sex-RARC and standard RARC cohorts.

### 2.4. Statistical Analysis

Descriptive analyses were used. Frequencies and proportions were reported for categorical variables. Medians and interquartile ranges (IQRs) were reported for continuously coded variables.

The Kaplan–Meier method was performed to report daytime and nighttime continence recovery probabilities. Continence rates were computed at 3, 6, 12 and 18 months after surgery.

Differences between questionnaires’ domains scores evaluated at the baseline 3-month, and 1-year follow-up were assessed with the Friedman test.

In the supplementary outcomes, comparison, continuous, and categorical variables were compared with a Student’s *t*-test and a chi-square test respectively. The Kaplan–Meier method was performed to compare daytime continence recovery probabilities between sex-RARC and standard RARC cohorts. Continence rates were computed at 3, 6, 12, and 18 months after surgery and the log-rank test was applied to assess any statistically significant differences between the two groups.

All *p*-values < 0.05 were considered statistically significant. Statistical analysis was performed using SPSS v24 (IBM Corp., Armonk, NY, USA).

## 3. Results

Baseline and clinical features were reported in Table 1. Median operative time was 255 min (IQR 250–399). The median hospital stay was 7 days (7–12). Low-grade Clavien complications occurred in four patients (36.3%) while high grade complications were not observed. Seven patients (63.7%) had an organ-confined disease at the pathologic specimen; nodal involvement and positive surgical margins were not detected in any case (Table 2).

All patients had a minimum follow-up period of one year. At a median follow-up of 28 months (IQR 14–51), no patient developed a new onset of chronic kidney disease stage 3b. One patient reported a neobladder stone formation, and one patient developed a ureteroileal anastomotic stricture and required robotic reimplantation 18 months following surgery (Table 2).

One-year daytime and nighttime continence recovery probability were 90.9% and 86.4%, respectively (Figure 1a,b). Three patients performed self-catheterization twice a day (early morning and before night rest).

Concerning the EORTC-QLQ-C30 questionnaire, global health status/quality of life, physical, and emotional functioning items improved significantly over time (all *p* ≤ 0.04), while no differences were observed in any other items evaluated (all *p* ≥ 0.10) (Appendix A, Figure 2).

According to the EORTC-QLQ-BLM30 questionnaire, specific for BCa, urinary symptoms worsened at three months with a significant recovery at one year (*p* = 0.02). Accordingly, when matching the baseline with 1-year scores, the values were comparable (*p* = 0.08) (Appendix A, Figure 2).

Finally, the FSFI global score and FSFI domains such as arousal, lubrication, orgasm, satisfaction, and pain worsened over the first three months with a subsequent improvement at one year (all *p* ≤ 0.04). Moreover, comparing baseline vs. 1-year scores, arousal and orgasm domains experienced a complete recovery (*p* = 0.10 and *p* = 0.10, respectively), while lubrication, satisfaction, and pain domains, as well as FSFI global scores, experienced a satisfying improvement but were statistically significantly lower than baseline (all *p* ≤ 0.025) (Appendix A, Figure 3). Overall, 8 out of 11 patients (72.7%) were sexually active at the 12-month evaluation.

As supplementary analysis, 36 standard RARC patients were compared with the sex-RARC cohort. The two cohorts were homogeneous for all baseline, clinical, and pathological features (all *p* ≥ 0.14) except for age, with sex-sparing patients being significantly younger (47.1 vs. 61.7 years, *p* < 0.001) (Appendix A).

Perioperative complications and hospital stay were comparable between groups (*p* = 0.25 and *p* = 0.67 respectively) (Appendix A).

With regard to functional outcomes, no significant differences were observed for the last estimated glomerular filtration rate (*p* = 0.43), neobladder stone formation rate (*p* = 0.93), and 1-year incidence of ureteroileal strictures (*p* = 0.67) (Appendix A). Daytime continence recovery probability was significantly higher in the sex-sparing cohort (1-year rate 90.9% vs. 74%, log-rank *p* = 0.02) (Appendix A).

## 4. Discussion

Functional outcomes among women undergoing RC have been poorly addressed in the literature [11]. Urinary function is the most studied issue, although daytime and nighttime continence rates range significantly across studies due to a heterogeneity of definitions for continence, different inclusion criteria, and a lack of questionnaire adoption, as these are omitted in most studies [11]. In addition, Zahran et al. conducted a systematic review aiming to evaluate female sexual dysfunction post RC and urinary diversion, considering it an important predictor of HRQoL post RC. According to the 11 studies included, the most frequently detected sexual disorders were loss of sexual desire and orgasm disorders (49% and 39%, respectively) [12]. Notwithstanding, the authors called for the use of standardized tools in order to properly assess the outcomes of this technique from the patients’ perspective and reported poor evidence from the available literature. Moreover, no data were available about RARC in females.

The concept of sex-RC in female patients was first introduced by the Bern team in 2007, when, in select female patients with an absence of invasive cancer at the level of the trigone or dorsolateral side walls of the bladder, they emphasized the functional advantages deriving from the preservation of the utero-vaginal hypogastric plexus, which is usually sacrificed in the standard procedure [7].

These results were corroborated by meticulous cadaveric studies elucidating topographic anatomic details of the nervous autonomic system in women, with their clinical nuances [13,14]. The SHP was identified as a single anatomical complex located below the aortic bifurcation, ventral to the sacral promontory. After the promontory, the SHP divides into right and left hypogastric nerves that more caudally plunge into the inferior pelvic IHP, composed by utero-vaginal, vesical, and rectal plexus.

As expected, preservation of these neural structures has an impact on recovery of urinary continence and on voiding function. Accordingly, data coming from gynecological studies report intrinsic sphincter deficiency resulting from hysterectomy as a consequence of urethral denervation after an extensive pelvic dissection [15,16]. Moreover, a pelvic autonomous nervous system affects all the domains of sexuality, such as sexual desire, arousal, lubrication, orgasm, satisfaction, and post-RC sexual dysfunction, often associated with pain disorders, such as dyspareunia, vulvodynia, and vaginismus, each being a consequence of autonomic and nociceptive nerve injuries, and a shortening or a narrowing of the vagina with an unavoidable negative impact on HRQoL [17].

In the literature, there are few existing series reporting sexual function results after sex-RC, all of them with an open approach and most of them with a small number of patients and without assessment of HRQoL through self-administered standardized questionnaires. Nandipati et al. focused on preservation of the lateral walls of the vagina, in which are embedded nervous fibers directed to the paraurethral tissue, involved in clitoral vascularization. In the small cohort of six women who underwent the sex-sparing approach, 12-month FSFI remained stable, while it declined in the standard RC group [18].

Furthermore, a significant improvement in all domains of the FSFI questionnaire has been reported in 13 sex-sparing RC patients evaluated at Mansoura Urology Department, with daytime and nighttime continence rates of 100% and 92%, respectively [19].

In this context, the EAU MIBCa Guideline Panel recently commissioned a systematic review aiming to assess the effect of sexual-function-preserving surgical techniques on outcomes in women receiving RC and ON substitution for BCa [5]. Sex-sparing approaches were found to be oncologically safe in well-selected patients, with sexual function appearing to be improved among those women undergoing gynecologic organ-preserving and nerve-sparing approaches. Nevertheless, most of the studies analyzed were retrospective and only contained a small number of patients [5]. Hence, according to EAU guidelines, data regarding sex RC in female patients are still considered immature and it is not yet considered a standard treatment, but an option to be taken into consideration for women highly motivated to preserve sexual function so long as strict oncologic inclusion criteria are met [1]. In addition, though the oncological equivalence of open and robotic RC has been extensively assessed, [20,21,22] and the robotic approach has been widely adopted in the male sex-sparing counterpart with excellent functional results [23], there are no reports on sex-RARC in female patients.

Hence, in this paper and in the accompanying video, we firstly described surgical steps of sex-RARC in female patients and reported perioperative and functional outcomes of our initial series with a minimum 1-year follow-up. In the video, we clearly highlight the differences with the standard technique with special attention paid to the preservation of the utero-vaginal component of the IHP. We strongly believe that robotic technology offers undebatable advantages in meticulously following and dissecting the appropriate surgical planes since IHP fibers are usually embedded in dense connective tissue, and consequently are not always easy to preserve. Despite the small cohort (11 patients), the excellent continence results (daytime 90.9% and nighttime 86.4% at one year) and the encouraging rate of sexually active patients (72.7% at one year) reflect the proper respect of the crucial anatomical structures and reinforce the efficacy of sex-RC in properly selected women. In addition, the oncological effectiveness with an absence of any recurrence corroborates our results.

Another important point of strength regarding the reliability of our results is our adoption of standardized questionnaires to assess the quality of life, urinary symptoms, and sexual activity (i.e., EORTC-C30, BLM30, and FSFI), which are rarely used in most studies. Moreover, the minimal invasiveness of the robotic approach represents a further issue to consider when considering young sexually and socially active women.

Furthermore, our technique may avoid devastating complications such as vaginal dehiscence and evisceration which have been reported after minimally invasive radical cystectomy [24,25].

Nevertheless, the present paper is not devoid of limitations. The small sample size, the strict inclusion criteria, and the need for advanced robotic surgical skills are significant limitations to a wide reproducibility of these outcomes in daily practice. Finally, BCa recurrence usually occurs within two years of radical cystectomy. In this respect, the follow-up duration might be inadequate.

## 5. Conclusions

In selected populations, sex-RARC-iN can be offered to female patients motivated to preserve sexual function as an oncologically safe procedure, associated with favorable functional outcomes. The meticulous anatomical preservation of utero-vaginal components of IHP represents the cornerstone of a quick and effective recovery of physiological functions in terms of urinary continence and sexual activity. A proper comparison of outcomes with the conventional RARC-iN technique requires properly designed prospective randomized trials.

## Figures and Tables

**Figure 1 jcm-09-00577-f001:**
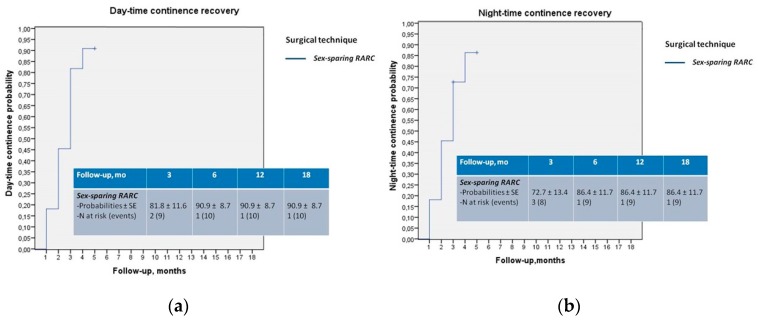
(**a**,**b**) Kaplan–Meier analysis reporting daytime and nighttime continence recovery probabilities.

**Figure 2 jcm-09-00577-f002:**
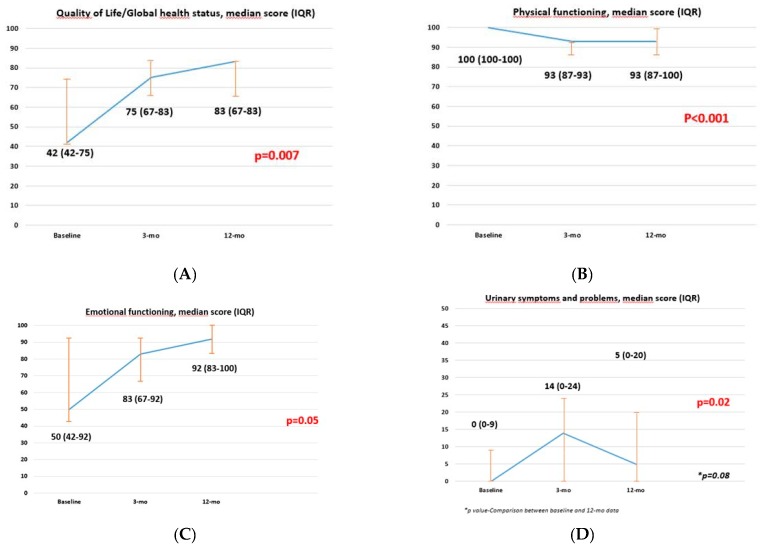
Graphs showing EORTC-QLQ-C30 and EORTC-QLQ-BLM30 questionnaire items displaying statistical significance according to the Friedman test. (**A**) Quality of Life/Global health status; (**B**) Emotional functioning; (**C**) Physical functioning; (**D**) Urinary symptoms and problems

**Figure 3 jcm-09-00577-f003:**
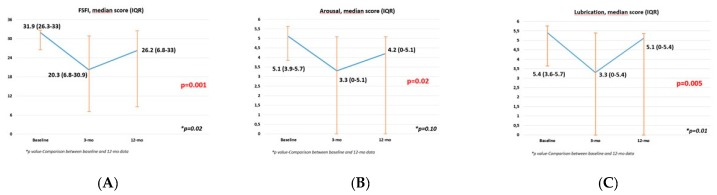
Graphs showing global Female Sexual Function Index (FSFI) and FSFI single domains questionnaire displaying statistical significance according to the Friedman test. (**A**) FSFI; (**B**) Arousal; (**C**) Lubrication; (**D**) Orgasm; (**E**) Satisfaction; (**F**) Pain.

**Table 1 jcm-09-00577-t001:** Baseline and clinical characteristics.

Patients, n 11	Sex-Sparing RARC
Age, year, mean (±SD)	47.1 (13)
BMI, mean (±SD)	23.1 (3.3)
ASA score, *n* (%)	
1	4 (36.4)
2	6 (54.6)
3	1 (9)
4	-
Preoperative eGFR, mL/min, mean (±SD)	84 (26.8)
Preoperative Hgb, g/dL, mean (±SD)	12.6 (1.9)
Neoadjuvant Chemotherapy, *n* (%)	4 (36.3)

**Table 2 jcm-09-00577-t002:** Perioperative, pathologic, oncologic and functional outcomes.

*Patients*	*Sex-Sparing RARC (11)*
Operative time, min, median (IQR)	255 (250–399)
Hgb at discharge, g/dL, median (IQR)	10.8 (9.1–11.9)
Hospital stay, days, median (IQR)	7 (7–12)
Complications, *n* (%)	4 (36.3)
*Clavien Low grade (1–2)*	4 (36.3)
*Clavien High grade (≥3)*	0 (0)
pT stage, *n* (%)	
0, a, is	6 (54.6)
1	1 (9.1)
2	-
3	4 (36.3)
4	-
pN stage, *n* (%)	
0	11(100)
1	-
2	-
Lymph node count, mean (±SD)	26.2 (14.3)
Positive surgical margins, *n* (%)	0 (0)
Follow-up, months, median (IQR)	28 (14–51)
1-Year recurrence-free survival, *n* (%)	11 (100)
1-Year cancer-specific survival, *n* (%)	11 (100)
1-Year overall survival, *n* (%)	11 (100)
Last eGFR, mL/min, mean (±SD)	79.2 (23.7)
Ureteroileal strictures, pts (%)	1 (9)
Neobladder stones, *n* (%)	1 (9)
Need for intermittent self-catheterization, *n* (%)	3 (27.2)

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
