# Peer review of "Sex-Sparing Robot-Assisted Radical Cystectomy with Intracorporeal Padua Ileal Neobladder in Female: Surgical Technique, Perioperative, Oncologic and Functional Outcomes"

_jcm, 2020, doi:10.3390/jcm9020577_

Round 1

Reviewer 1 Report

I have read the authors' response. They clarified some statements and I am satisfied.

Author Response

Reviewer 1

I have read the authors' response. They clarified some statements and I am satisfied.

Thanks for this comment.

Reviewer 2 Report

The author described the technique of Sex-sparing Robot assisted Radical Cystectomy with intracorporeal neobladder in female patients. This is a well written and interesting in the field of minimally invasive urologic surgery. I have found a few issues that, once addressed, will improve the manuscript.

In this paper, you only showed the data about cases received Sex-sparing Robot assisted Radical Cystectomy with intracorporeal neobladder. Could you show the preoperative and postoperative data about cases with non-organ preserved radical cystectomy (anterior pelvic exenteration) and compared data between sex-sparing robot assisted radical cystectomy with intracorporeal neobladder group and non-organ preserved radical cystectomy group?

Bladder cancer recurrence usually occurs within 2 years after radical cystectomy. In this respect, the follow-up duration is inadequate. Please refer to this point in discussion part.

Recently, vaginal dehiscence and evisceration after minimally invasive radical cystectomy has been reported. Your technique can avoid this devastating complication. I recommend you should refer to this merit in your technique in discussion part.

Lin FC, Medendorp A, Van Kuiken M, Mills SA, Tarnay CM. Vaginal Dehiscence and Evisceration after Robotic-Assisted Radical Cystectomy: A Case Series and Review of the Literature. Urology. 2019.

Kanno T, Ito K, Sawada A et al. Complications and reoperations after laparoscopic radical cystectomy in a Japanese multicenter cohort. Int J Urol. 2019.

Author Response

Reviewer 2

The author described the technique of Sex-sparing Robot assisted Radical Cystectomy with intracorporeal neobladder in female patients. This is a well written and interesting in the field of minimally invasive urologic surgery. I have found a few issues that, once addressed, will improve the manuscript.

 In this paper, you only showed the data about cases received Sex-sparing Robot assisted Radical Cystectomy with intracorporeal neobladder. Could you show the preoperative and postoperative data about cases with non-organ preserved radical cystectomy (anterior pelvic exenteration) and compared data between sex-sparing robot assisted radical cystectomy with intracorporeal neobladder group and non-organ preserved radical cystectomy group?

Thanks for this interesting comment. As already highlighted in our conclusions, a proper comparison of outcomes with conventional RARC-iON technique would require properly designed prospective randomized trials; accordingly, a retrospective comparison of the two cohorts might be disomogeneous in terms of age, comorbidity, clinical T stage, with intrinsic selection bias.

Moreover it is not the aim of this paper, therefore EORTC QLQ-C30, EORTC QLQ-BLM30 and FSFI questionnaires are not available for the standard RARC cohort.

Nevertheless, as requested by the reviewer, we provided supplementary data concerning preoperative and postoperative data of standard RARC-iON, with outcomes comparison with Sex-sparing cohort.

Experimental section was amended as follows:

“As supplementary data, we reported preoperative, perioperative pathologic  and functional characteristics comparison of sex-RARC and standard RARC cohorts. “

“In the supplementary outcomes comparison, continuous and categorical variables were compared with Student t and chi-square test, respectively. Kaplan-Meier method was performed to compare day-time continence recovery probabilities between sex-RARC and standard RARC cohorts. Continence rates were computed at 3,6,12 and 18 months after surgery and the log-rank test was applied to assess statistical significance between two groups.”

Results section was amended as follows:

“As supplementary analysis, 36 standard RARC patients were compared with the sex-RARC cohort. The two cohorts were homogeneous for all baseline, clinical and pathologic features (all p≥0.14), except for age, being sex-sparing patients significantly younger (47.1 vs 61.7 yrs, p<0.001) (Supplementary Table 4 and Supplementary Table 5).

Perioperative complications and hospital stay were comparable between groups (p=0.25 and p=0.67, respectively) (Supplementary Table 5).

With regard to functional outcomes, no significant differences were observed for last estimated glomerular filtration rate (p=0.43), neobladder stone formation rate (p=0.93) and 1-yr incidence of ureteroileal strictures (p=0.67) (Supplementary Table 5). Day-time continence recovery probability was significantly higher in Sex-sparing cohort (1-yr rate 90.9% vs 74%, log-rank p=0.02) (Supplementary Figure 1). “

Bladder cancer recurrence usually occurs within 2 years after radical cystectomy. In this respect, the follow-up duration is inadequate. Please refer to this point in discussion part.

Thanks for this comment. The Discussion section was amended as follows:   “Finally, BCa recurrence usually occurs within 2 years after radical cystectomy. In this respect, the follow-up duration might be inadequate. “

Recently, vaginal dehiscence and evisceration after minimally invasive radical cystectomy has been reported. Your technique can avoid this devastating complication. I recommend you should refer to this merit in your technique in discussion part.

Lin FC, Medendorp A, Van Kuiken M, Mills SA, Tarnay CM. Vaginal Dehiscence and Evisceration after Robotic-Assisted Radical Cystectomy: A Case Series and Review of the Literature. Urology. 2019.

Kanno T, Ito K, Sawada A et al. Complications and reoperations after laparoscopic radical cystectomy in a Japanese multicenter cohort. Int J Urol. 2019.

Thanks for this comment. The Discussion section was amended as follows: “Furthermore, our technique may avoid devastating complications, such as vaginal dehiscence and evisceration, which were reported after minimally invasive radical cystectomy [24, 25].

The two references suggested by the reviewer were added to our reference list as number 24 and 25.

Round 2

Reviewer 2 Report

The authors well addressed all the issues that reviewers pointed out.